# Ebola incidence and mortality before and during a lockdown: The 2022 epidemic in Uganda

Jonathan Izudi[1,2]*, Allan Komakech[3,4], Job Morukileng[3], Francis Bajunirwe[1]

**1** Department of Community Health, Mbarara University of Science and Technology, Mbarara, Uganda, **2** Infectious Diseases Institute, Makerere University College of Health Sciences, Kampala, Uganda, **3** Uganda National Institute of Public Health, Ministry of Health, Kampala, Uganda, **4** Institute of Public Health and Management, Clarke International University, Kampala, Uganda

* jonahzd@gmail.com

**Data Availability Statement:** All relevant data are within the manuscript and its Supporting Information files.

## Abstract

On September 20, 2022, an Ebola Disease (EBOD) outbreak was declared in Mubende district, Central Uganda. Following a rapid surge in the number of cases and mortality, the Government of Uganda imposed a lockdown in the two most affected districts, Mubende and Kassanda. We describe the trends in EBOD incidence and mortality nationally and in the two districts before and during the lockdown and the lessons learned during the epidemic response. We retrieved data from the Ministry of Health situation reports from September 20, 2022, when the EBOD outbreak was declared until November 26, 2022, when the lockdown ended. We graphed trends in EBOD morbidity and mortality during a 3-week and 6-week lockdown, computed the EBOD case fatality rate, and summarized the major lessons learned during the epidemic response. We found case fatality rate during the pre-lockdown, 3-week lockdown, and 6-week lockdown period was 37.9% (22/58), 39.3% (53/135), and 38.7% (55/142), respectively. In the early weeks of the lockdown, EBOD incidence and mortality increased nationally and in Kassanda district while Mubende district registered a decline in incidence and stagnation in mortality. With the extension of the lockdown to six weeks, the EBOD incidence and mortality during the 4-6-week lockdown declined compared to the pre-lockdown period. In conclusion, the EBOD incidence and mortality remained higher in the early weeks of the lockdown than during the pre-lockdown period nationally and in one of the two districts. With extended lockdown, incidence and mortality dropped in the 4-6-week period than the pre-lockdown period. Therefore, reliance on known public health measures to control an EBOD outbreak is important.

## Introduction

On September 20, 2022, the Uganda Ministry of Health (MoH) confirmed a case of Ebola Disease (EBOD) outbreak caused by the Sudan strain [1]. It is the seventh epidemic caused by this specific strain since 1976, with four epidemics recorded in Uganda and three in Sudan. The

**Funding:** The author(s) received no specific funding for this work.

**Competing interests:** The authors have declared that no competing interests exist.

last epidemic due to the Sudan strain was in Uganda in 2012 [2]. The index case in the 2022 EBOD epidemic was a 24-year-old male(1), a resident of Madudu sub-county in Mubende District in central Uganda, who later presented at Mubende Regional Referral Hospital with high-grade fever, convulsions, blood-stained vomitus, and diarrhea [1].

EBOD is lethal and the case fatality rate (CFR) varies by species. Our latest systematic review and meta-analysis of EBOD data from 1976 to 2022 (the most updated) show that the EBOD CFR varies from 51.6% to 69.4%, averaging at 60.6% [3]. The CFR is highest for the Zaire ebolavirus at 66.6% (95% CI 55.9–76.8) followed by Sudan ebolavirus at 48.5% (95% CI 38.6–58.4), then Bundibugyo ebolavirus at 32.8% (95% CI 25.8–40.2), and lowest for Tai Forest virus at 0% (95% CI 0–97.5%). To prevent the spread of the EBOD outbreak, the MOH instituted immediate response measures: activating the National Task Force and the Incident Management Team and repurposing the COVID-19 Incident Command to the EBOD outbreak response in the affected district [4]. Response measures at the district level included surveillance, risk communication, deployment of the National Response Team to support the epidemiological investigation, contact tracing, and case finding. Additionally, on-site laboratory testing for EBOD at Mubende Regional Referral Hospital with a sample turn-around time of 6 hours, social mobilization, infection prevention and control, and logistical support was initiated [5]. Despite these measures, the EBOD continued to spread within and beyond Mubende district to several neighboring districts of Kassanda, Kyegegwa, Kagadi, and Bunyangabu and some far off like Wakiso and Kampala.

On October 15, 2022, the President of the Republic of Uganda imposed a 3-week lockdown on the two most affected districts of Mubende and Kassanda [5]. Effective implementation of the lockdown started on October 16, 2022, and ended on November 5, 2022. However, on November 5, 2022, the lockdown was extended by another three weeks [6]. Such lockdowns have recently been used successfully in the control of COVID-19 spread [7–10]. Evidence suggests that controlling population movement patterns within and between areas with and without an outbreak can limit the spread of communicable diseases thus necessitating the inclusion of population movements in preparedness and response strategies [10].

In 2015, Sierra Leone imposed a 3-day national lockdown to contain an EBOD outbreak, dramatically reducing population mobility [11]. In Uganda, this is the first time a lockdown has been enforced to control an EBOD outbreak. We described trends in EBOD incidence and mortality nationally and in the two most affected districts in Uganda before and during the lockdown and the lessons learned during the epidemic response. This evidence is important in informing future public health decisions regarding EBOD control in Uganda and other countries at risk for EBOD epidemics.

## Methods

### Study design, setting, and data sources

We conducted a retrospective review of available records. The sources included online EBOD situation reports that were published by the Uganda MoH between September 20, 2022, and November 26, 2022. We abstracted data (S1 Data) on EBOD incidence and mortality for the entire country, and separately for the two districts under a lockdown, namely Mubende and Kassanda in central Uganda. EBOD situation reports were produced daily to provide an update on EBOD incidence and mortality, recoveries, contacts traced, places affected, and public health response.

### Description of lockdown

The lockdown included movement restrictions for people, vehicles, and motorcycle taxis both in and out of Mubende and Kassanda districts, with only cargo trucks allowed to cross the districts but without stopping. Places of public gathering like churches, mosques, markets, entertainment venues, bars, and nightclubs were all closed but within the district, movements remained unrestricted although they slowed. Night-time curfews supervised by security forces of the Uganda Police and Army were imposed from 7.00 pm to 6.00 am local time [5]. The security forces arrested persons who faulted the lockdown measures. Offices at public and private workplaces were ordered closed and only businesses selling essential items such as food and medicines were allowed to operate. All primary and secondary schools in the two districts were closed. When six school children were diagnosed with EBOD in Kampala in late October 2022, primary and secondary schools nationally closed two weeks earlier than had been scheduled to prevent further spread.

The period September 20, 2022, to October 15, 2022 (26 days) was the pre-lockdown period, October 16, 2022, to November 5, 2022, was the 3-week lockdown period (21 days), and the 6-week lockdown was October 16, 2022, to November 26, 2022 (42 days).

### Outcomes

The outcome of interest was EBOD incidence and mortality at the national level and for each of the two most affected districts. All EBOD incident cases were confirmed by laboratory testing using Deoxyribonucleic acid-polymerase chain reaction or DNA-PCR at the Uganda Virus Research Institute Viral Hemorrhagic Fever Laboratory and subsequent deaths of these cases were attributed to EBOD.

### Statistical analysis

We performed a descriptive analysis in Stata version 15. Here, we summarized the EBOD case counts during the pre-lockdown, 3-week, and 6-week lockdown periods, and calculated the respective CFRs for each period. The CFRs were computed as the percentage of the total number of EBOD confirmed deaths over the total number of confirmed EBOD incident cases. We plotted the daily EBOD incident cases and mortality over time to assess trends.

### Human subjects' issues

No ethical approval was required since the data analyzed are de-identified, publicly accessible, and were provided to the public in an aggregated format regularly by the Uganda MoH.

### Inclusivity in global research

Additional information regarding the ethical, cultural, and scientific considerations specific to inclusivity in global research is included in the Supporting Information (S1 File).

## Results

### EBOD case counts nationally and in the most affected districts before and during the lockdown

Table 1 summarizes the EBOD case counts during the pre-lockdown and lockdown periods. Compared to the pre-lockdown case counts, more EBOD cases were reported in the first 1–3 weeks of the lockdown compared to the pre-lockdown period at the national level (77 versus 55) and Kassanda district (42 versus 4). However, a twofold decline in EBOD incidence was

**Table 1. EBOD incidence and mortality case counts during a 3-week and 6-week lockdown at the national level and in the two most affected districts.**

| | EBOD incidence case counts | | | |
|---|---|---|---|---|
| Setting | Pre-lockdown | 1-3-week lockdown | 4-6-week lockdown | All (1–6 weeks) |
| National | 58 | 77 | 7 | 84 |
| Mubende | 46 | 13 | 1 | 14 |
| Kassanda | 4 | 42 | 3 | 45 |
| | EBOD mortality case counts | | | |
| Setting | Pre-lockdown | 1-3-week lockdown | 4-6-week lockdown | All (1–6 weeks) |
| National | 22 | 31 | 2 | 33 |
| Mubende | 20 | 20 | 1 | 21 |
| Kassanda | 0 | 8 | 0 | 8 |

observed in Mubende district (13 versus 46). During the 4–6 weeks of the lockdown, the EBOD cases dropped at all levels when compared to the pre-lockdown period nationally (7 versus 58) and in Mubende (1 versus 46) and Kassanda (3 versus 4) districts.

## EBOD mortality nationally and in the most affected districts before and during the lockdown

Compared to the pre-lockdown mortality cases, the number of mortality cases in the first 1–3 weeks of the lockdown was higher at the national level (31 versus 22) and in Kassanda district (8 versus 0) but it remained the same in Mubende district during both periods (n = 20). However, mortality cases in the 4–6 weeks of the lockdown were fewer compared to the pre-lockdown period at the national level and in the two districts under a lockdown. The case fatality rate (CFR) during the pre-lockdown, 1-3-week lockdown, 4–6 week, and 1-6-week lockdown periods were 37.9% (22/58), 39.3% (53/135), 28.6% (2/7), and 38.7% (55/142), respectively (p = 0.862, p = 0.628, and p = 0.928 respectively). The trends in EBOD incidence and mortality are graphically presented at the national level (Fig 1). Fig 1 National EBOD incident cases and deaths in Uganda before and during the lockdown in 2022. Trends in EBOD incidence and mortality in Mubende district is shown in Fig 2. Fig 2 EBOD incident cases and deaths in Mubende district before and during the lockdown in 2022. Fig 3 shows the trends in EBOD incidence and deaths in Kassanda district. Fig 3 EBOD incident cases and deaths in Kassanda district before and during the lockdown in 2022.

## Discussion

This is the first time Uganda implemented a lockdown to control the EBOD spread after four previous successful control efforts [12]. We found an increase in EBOD incidence and mortality in the early weeks of the lockdown compared to the pre-lockdown period at the national level and in Kassanda district, with Mubende district recording a decline in incidence and a stagnation in mortality. With the extension of the lockdown to six weeks, we found the EBOD incidence and mortality during the 4-6-week lockdown period declined compared to the pre-lockdown period.

Our study has some strengths and limitations. To the best of our knowledge, this is the first study to report the trends in EBOD incidence and mortality following a lockdown nationally and globally. In Sierra Leone, a 3-day lockdown was enforced in 2015 [11] but the trends in EBOD incidence and mortality following the lockdown were not described due to the short time. We retrieved and analyzed data from the MOH situation report, which is a reliable data source.

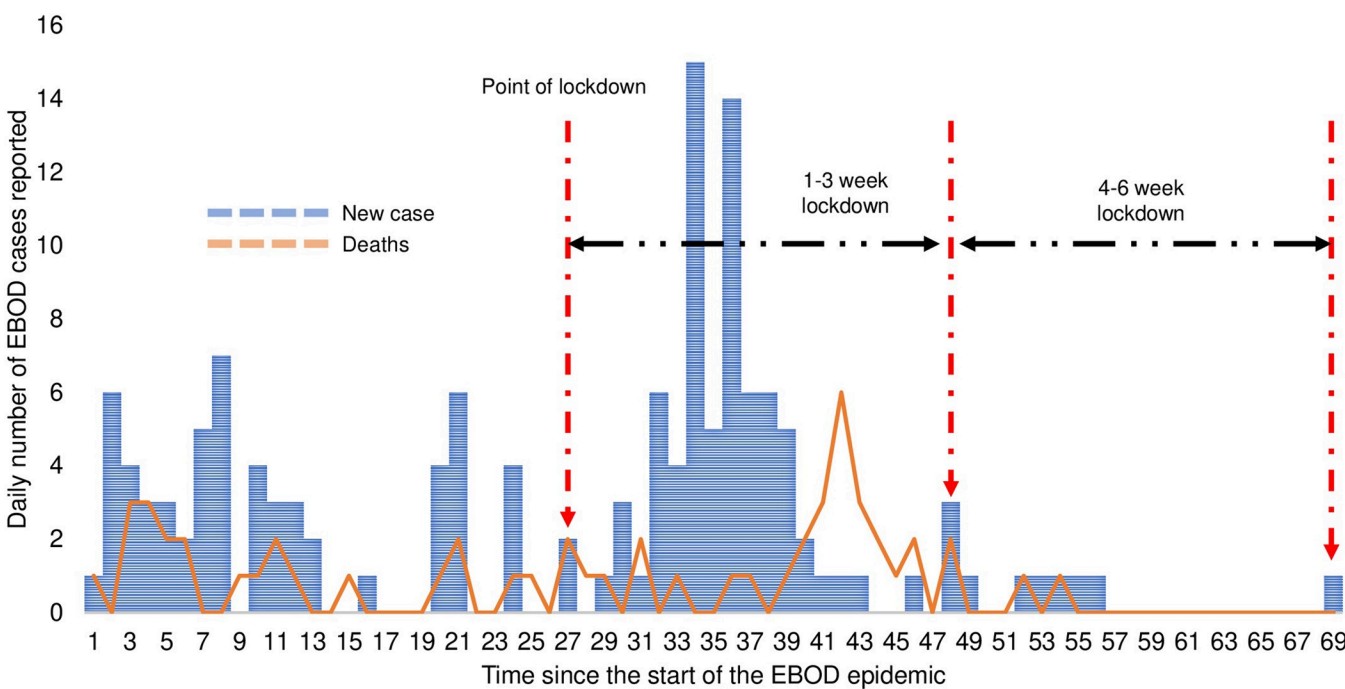

**Fig 1.**

Limitations of the study include a lack of an Ebola Treatment Unit in one of the districts, Kassanda district, hence all its incident EBOD cases were referred to Mubende Regional Referral Hospital Ebola Treatment unit for management. This might have obscured the realization

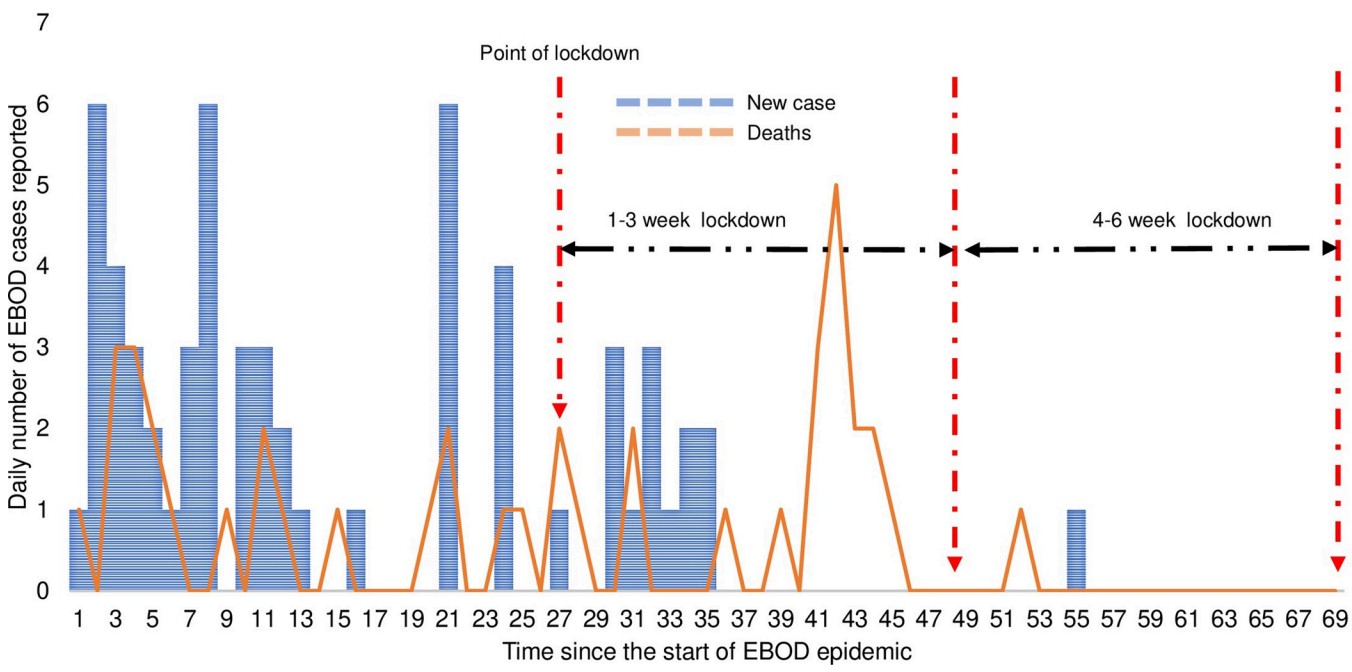

**Fig 2.**

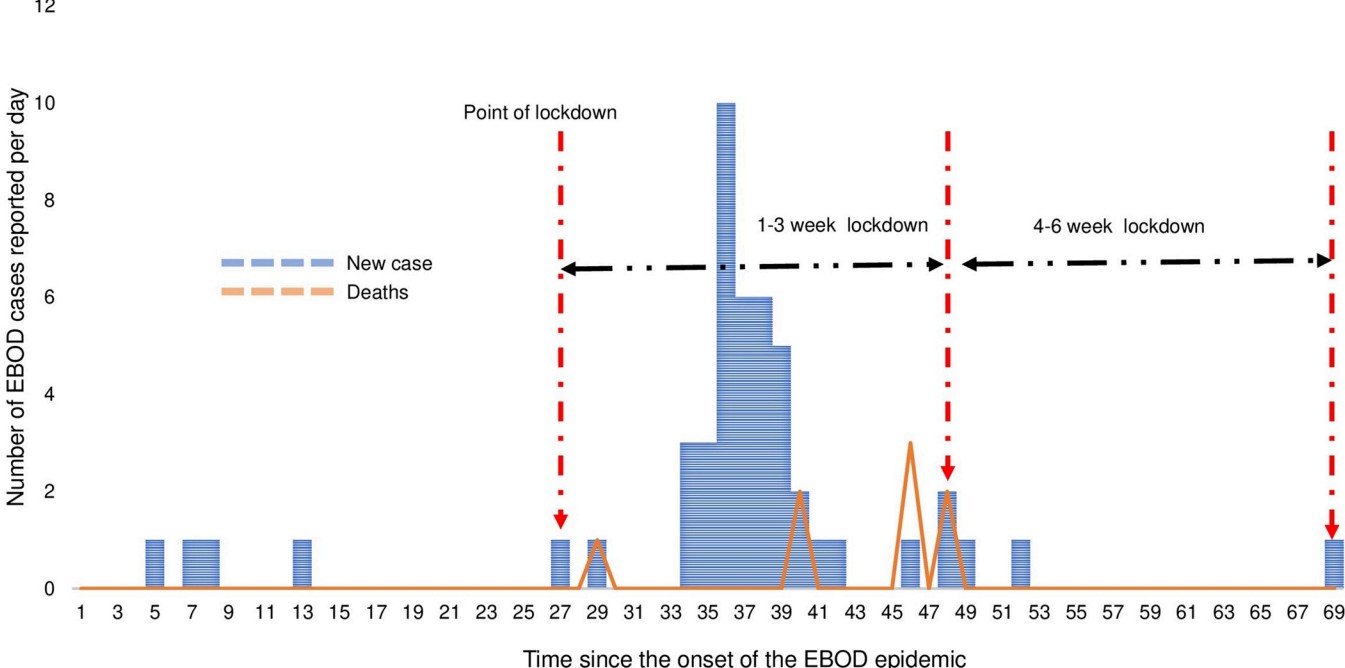

**Fig 3.**

of the impact of the lockdown on EBOD mortality in Mubende district. The Uganda MoH implemented a standard EBOD prevention and control package but the intensity of the implementation might have varied between the districts. This variation was difficult to measure and quantify. We do not have data on population movements within the districts and whether the EBOD cases were imported across the districts or not.

We found comparable CFRs between the pre-lockdown and the 1–3 week and 1-6-week lockdown periods. We present a few potential explanations for these findings. First, some contacts flouted the lockdown measures by traveling to other districts and yet eventually became cases hence spreading the EBOD. Second, sociocultural barriers deterred EBOD cases from seeking clinical care and the population from conforming to public health preventive measures. We noted that a few EBOD cases feared seeking care at Mubende Regional Referral Hospital citing possible loss of vital body organs at the hands of crooked healthcare workers as rumors of illegal organ removal had circulated [13]. Some of the early EBOD cases sought care from private health facilities and traditional or complementary medicine practitioners like traditional healers leading to exposure of several contacts to the Ebola virus. This is consistent with an earlier finding in West Africa where sociocultural barriers played a significant role in the persistent spread of EBOD [14]. Third, the MoH used home-based isolation of EBOD contacts and this might have potentially exposed some persons in the community including household/or family members, friends, and relatives to the Ebola virus. Some of the incident cases had likely established contacts in the community. Although isolation is standard practice, it was unclear whether the practice was supervised or whether there were incentives to ensure adherence.

Our finding of an increase in EBOD incidence and mortality in the early weeks of the lockdown requires cautious interpretation. The lockdown is not expected to have an immediate impact. No prior evidence is available to support the effectiveness of a lockdown in decreasing the Ebola spread and therefore mortality. However, our biologically plausible viewpoint is that

the rise in Ebola incidence and mortality in the early weeks of the lockdown suggests several persons in the community had already been exposed to the Ebola virus prior to the lockdown, with many becoming cases thereafter.

This argument is supported by surveillance data that show probable EBOD cases appeared around August 7, 2022, or potentially earlier [15]. Therefore, even before the EBOD outbreak was declared, some cases may have spread across several neighboring and perhaps distant districts like Kampala, making the immediate EBOD outbreak control difficult. Accordingly, the immediate worsening of EBOD incidence and mortality might reflect a spillover of EBOD cases that were exposed during the pre-lockdown period.

## Lessons learned during the EBOD epidemic response and the lockdown

In the past, Uganda successfully controlled EBOD outbreaks without enforcing a lockdown but relied on strong coordination at the national, district, and community levels, rapid outbreak response, timely risk communication, community-based surveillance, and community engagement among others [12,16]. Our analysis does not provide evidence as to whether a lockdown, when combined with these well known traditional measures creates a more efficient approach to control the spread of the infection. Lockdowns are generally challenging to implement. Experiences in the affected community indicate that the lockdown was not completely adhered to. For example, a man from one of the affected districts who had contact with a case of EBOD escaped and traveled to Kampala where he became ill and died but that was before he infected over a dozen other people, including his children who attended three different schools. Elsewhere, a recent study in Uganda demonstrates that communities are closely interconnected for social and economic reasons leading to non-adherence to guidelines that limit travel during an EBOD outbreak [17]. The large EBOD epidemic in West Africa teaches us important lessons that Uganda can pick from to control future epidemics. Harnessing community trust and creatively engaging grass root community entities is critical in epidemic control [18]. In Liberia [19], large numbers of active case finders were enlisted, households with contacts received supplies of food and water, and religious leaders including imams were tapped to lead response and burial teams. Health authorities in an unusual turn provided illegal drugs to gangs and food supplies to robbers to trace their contacts. Such unorthodox or otherwise creative solutions may be key emphasis in controlling community spread.

Our findings regarding the increasing EBOD incidence and mortality despite a lockdown also potentially point to challenges encountered by the healthcare systems in the control of EBOD in sub-Saharan Africa [20].

Inadequacies in health system preparedness and response have been reported at lower-level health facilities where there are budget constraints and a lack of rapid response teams [21]. Besides a low suspicion index for EBOD among healthcare workers, shortages of personal protective equipment and detergents have repetitively been reported during the 2022 EBOD outbreak response in Uganda. This might have stifled swift response leading to the infection of 18 healthcare workers with seven reported dead as of November 5, 2022 [22]. Also, the two districts encountered community-level challenges, namely stigmatization and discrimination against Ebola patients and their contacts, myths, and misconceptions about EBOD, low EBOD risk perception, and resistance to the evacuation of probable cases, including isolation of contacts. These sociocultural barriers are not unique and have been previously reported elsewhere in sub-Saharan Africa [20]. Effective treatment for EBOD using immunoglobulins [23] and vaccination exists just for the Zaire ebolavirus but access remains a challenge in low-income countries [24]. Accordingly, early diagnosis and prompt supportive care remain important options for preventing substantial morbidity and mortality due to other EBOD strains such as

the Sudan ebolavirus strain to increase survival and limit the spread of EBOD [20]. We emphasize that an EBOD outbreak response should focus on the timely identification of incident cases and linkage to clinical care, strict isolation, and adherence to infection control practices. In addition, ensuring a strong surveillance system with timely response, contact tracing, early risk communication, establishing a strong community-led health education system, and engaging the community leadership in designing, planning, monitoring, and evaluating epidemic response to counter sociocultural barriers are fundamental. Lastly, it is important to build the capacity of healthcare workers to enhance their level of clinical suspicion.

## Conclusion and recommendation

The 2022 EBOD outbreak in Uganda was characterized by high morbidity and mortality and continued to spread despite a 6-week lockdown. The CFRs were comparable between the pre-lockdown and the lockdown phases both nationally and in the two most affected districts. There is a need to strengthen the health system for epidemic preparedness and use lessons from past epidemics to creatively engage community entities like religious leaders, health teams, and volunteers in education, contact tracing, isolation, and burials during epidemic control.

## Supporting information

**S1 Data. Dataset.**
(DTA)

**S1 File. Inclusivity in global research.**
(DOCX)

## Acknowledgments

We are greatly indebted to the Uganda Ministry of Health for making the EBOD data publicly accessible.

## Author Contributions

**Conceptualization:** Jonathan Izudi, Job Morukileng, Francis Bajunirwe.

**Data curation:** Jonathan Izudi, Allan Komakech, Job Morukileng, Francis Bajunirwe.

**Formal analysis:** Jonathan Izudi, Francis Bajunirwe.

**Investigation:** Jonathan Izudi, Allan Komakech, Job Morukileng, Francis Bajunirwe.

**Methodology:** Jonathan Izudi, Allan Komakech, Job Morukileng, Francis Bajunirwe.

**Project administration:** Job Morukileng.

**Resources:** Allan Komakech, Francis Bajunirwe.

**Software:** Jonathan Izudi, Francis Bajunirwe.

**Supervision:** Francis Bajunirwe.

**Validation:** Jonathan Izudi, Job Morukileng, Francis Bajunirwe.

**Visualization:** Jonathan Izudi, Allan Komakech, Francis Bajunirwe.

**Writing – original draft:** Jonathan Izudi, Allan Komakech, Job Morukileng, Francis Bajunirwe.

**Writing – review & editing:** Jonathan Izudi, Allan Komakech, Job Morukileng, Francis Bajunirwe.

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
