## [Decision Letter · Decision Letter 0]

31 Oct 2023

PGPH-D-23-01494

Ebola incidence and mortality before and during a lockdown: the 2022 epidemic in Uganda

Dear Dr. Izudi,

Thank you for submitting your manuscript to PLOS Global Public Health. After careful consideration, we feel that it has merit but does not fully meet PLOS Global Public Health’s publication criteria as it currently stands. Therefore, we invite you to submit a revised version of the manuscript that addresses the points raised during the review process.

We look forward to receiving your revised manuscript.

Kind regards,

Wilber Sabiiti

Academic Editor

Journal Requirements:

1. Please send a completed 'Competing Interests' statement, including any COIs declared by your co-authors. If you have no competing interests to declare, please state "The authors have declared that no competing interests exist". 

2. We do not publish any copyright or trademark symbols that usually accompany proprietary names, eg (R), (C), or TM  (e.g. next to drug or reagent names). Please remove all instances of trademark/copyright symbols throughout the text, including © on References section.

Additional Editor Comments (if provided):

Dear Authors

After careful consideration of comments from peer review, you are asked to review the paper by addressing the reviewers' comments. A written permission from Ugandan Ministry of Health allowing you to publish the data will be required before this paper is considered for publication.

Reviewers' comments:

Reviewer's Responses to Questions

**Comments to the Author**

1. Does this manuscript meet PLOS Global Public Health’s publication criteria? Is the manuscript technically sound, and do the data support the conclusions? The manuscript must describe methodologically and ethically rigorous research with conclusions that are appropriately drawn based on the data presented.

Reviewer #1: Yes

Reviewer #2: Yes

2. Has the statistical analysis been performed appropriately and rigorously?

Reviewer #1: Yes

Reviewer #2: Yes

3. Have the authors made all data underlying the findings in their manuscript fully available (please refer to the Data Availability Statement at the start of the manuscript PDF file)?

Reviewer #1: Yes

Reviewer #2: Yes

4. Is the manuscript presented in an intelligible fashion and written in standard English?

Reviewer #1: Yes

Reviewer #2: Yes

5. Review Comments to the Author

Reviewer #1: In the article titled ‘Ebola incidence and mortality before and during a lockdown: the 2022 epidemic in Uganda’, Izudi J et al assess the incidence of cases and mortality peri-lockdown during the 2022 SUDV outbreak in Uganda. I found the paper to be generally well written and the methods appropriate.

I, however, hold the two concerns below:

Major:

The authors need to inform the Uganda MOH leadership about their utilization of the data in this manner; either via the Director General or Strategic Management committee. This should easily be done by sharing a summary of the write up of results or asking for a chance to present these findings as a dissemination activity. When completed, that dissemination activity can then be noted in the paper. This will override the social controversy of use of public epidemic data in absence of awareness or permission from the owner. While it’s not unethical; this data surrounds human subjects and we need to show social responsibility to the authorities and responders.

Minor

The conclusion should be restricted to inferences from the results. Any other insights should clearly be given as recommendations.

As it stands now, for example, the abstracts conclusion is extended to a recommendation that is not inferred from the results of the study.

I argue that any recommendation should be taken out of the the abstract and left for main body of the manuscript to avoid confusion

Conclusion ‘Overall, the EVD incidence and mortality remained higher in the early weeks of the lockdown than during the pre-lockdown period nationally and in one of the two districts. With extended lockdown, incidence and mortality dropped in the 4-6-week period than the pre-lockdown period.

Recommendations ( outside direct inference of results; but conceivable; only in body of MS) Therefore, strengthening the health system for epidemic preparedness and utilizing the lessons from past epidemics to creatively engage community entities like religious leaders, health teams, and volunteers in education, contact tracing, isolation, and burials during epidemic control might be important in controlling the EVD

Reviewer #2: This paper would been best presented as short article. The authors also need to sate whether the decrease in cases would be a natural epidemiological trend or the lock downs.. restrictions imposed actually helped slow down the process.

What are the actual lessons learnt learnt that the government of Uganda and other governments in L MICS can learn from to be able to combat epidemic.

6. PLOS authors have the option to publish the peer review history of their article (what does this mean?). If published, this will include your full peer review and any attached files.

**Do you want your identity to be public for this peer review?** For information about this choice, including consent withdrawal, please see our Privacy Policy.

Reviewer #1: **Yes: **Misaki WAYENGERA

Reviewer #2: No

---

## [Editor Report · Decision Letter 1]

28 Nov 2023

Ebola incidence and mortality before and during a lockdown: the 2022 epidemic in Uganda

PGPH-D-23-01494R1

Dear Dr Izudi

We are pleased to inform you that your manuscript 'Ebola incidence and mortality before and during a lockdown: the 2022 epidemic in Uganda' has been provisionally accepted for publication in PLOS Global Public Health.

Best regards,

Wilber Sabiiti

Academic Editor

No further comments.